# Thrombolysis Outcomes in Acute Ischaemic Stroke Patients with Pre-Existing Cognitive Impairment

**DOI:** 10.3390/life13041055

**Published:** 2023-04-20

**Authors:** Isabela V. P. Ramnarine, Omar W. Rasheed, Peter J. Laud, Arshad Majid, Kirsty A. Harkness, Simon M. Bell

**Affiliations:** 1Sheffield Institute of Translational Neuroscience, University of Sheffield, Glossop Road, Sheffield S10 2GF, UK; 2Statistical Services Unit, University of Sheffield, Sheffield S10 2HQ, UK; 3Sheffield Teaching Hospitals NHS Foundation Trust, Royal Hallamshire Hospital, Glossop Road, Sheffield S10 2GJ, UK

**Keywords:** thrombolysis, cognitive impairment, stroke, haemorrhage, ischaemic

## Abstract

Background: Thrombolysis treatment for ischaemic stroke in patients with pre-existing disabilities, including cognitive impairment, remains controversial. Previous studies have suggested functional outcomes post-thrombolysis are worse in patients with cognitive impairment. This study aimed to compare and explore factors contributing to thrombolysis outcomes, including haemorrhagic complications, in cognitively and non-cognitively impaired patients with ischaemic stroke. Materials and Methods: A retrospective analysis of 428 ischaemic stroke patients who were thrombolysed between January 2016 and February 2021 was performed. Cognitive impairment was defined as a diagnosis of dementia, mild cognitive impairment, or clinical evidence of the condition. The outcome measures included morbidity (using NIHSS and mRS), haemorrhagic complications, and mortality, and were analysed using multivariable logistic regression models. Results: The analysis of the cohort revealed that 62 patients were cognitively impaired. When compared to those without cognitive impairment, this group showed worse functional status at discharge (mRS 4 vs. 3, *p* < 0.001) and a higher probability of dying within 90 days (OR 3.34, 95% CI 1.85–6.01, *p* < 0.001). A higher risk of a fatal ICH post-thrombolysis was observed in the cognitively impaired patients, and, after controlling for covariates, cognitive impairment remained a significant predictor of a fatal haemorrhage (OR 4.79, 95% CI 1.24–18.45, *p* = 0.023). Conclusions: Cognitively impaired ischaemic stroke patients experience increased morbidity, mortality, and haemorrhagic complications following thrombolytic therapy. However cognitive status is not independently predictive of most outcome measures. Further work is required to elucidate contributing factors to the poor outcomes observed in these patients and help guide thrombolysis decision-making in clinical practice.

## 1. Introduction

Thrombolytic therapy has now become a standard treatment in cases of acute ischaemic stroke presenting within a 4.5 h window. Although thrombolytic therapy has been shown to have a net benefit in patients with ischaemic stroke, the associated risks of intracerebral haemorrhage (ICH) and early death are significant [1,2]. There is ambiguity surrounding thrombolysis administration in patients with pre-existing cognitive impairment [3,4,5]. Cognitively impaired patients present with more severe strokes, and often exhibit poorer outcomes following thrombolytic therapy [6,7]. These poorer outcomes include a higher risk of morbidity on discharge and higher risks of mortality, both within admission and within 6 months of follow-up [8,9].

Although the landmark study, the third International Stroke Trial (IST-3), did not have an age limit on participant recruitment, patients who were “already dependent in activities of daily living before acute stroke” were excluded [10]. Loss of the ability to perform activities of daily living is a defining feature of dementia; therefore, it can be assumed these patients were not represented in the trial. This suggests limited information exists about the risks and benefits of thrombolysing people with pre-stroke cognitive impairment.

It has been suggested that the underlying pathophysiology of conditions known to cause cognitive impairment is behind the poorer outcomes post-stroke [11]. These include cerebral amyloid angiopathy (CAA), which can be a disease entity on its own. CAA can also be present as part of Alzheimer’s disease and hypertensive angiopathy, which can lead to vascular dementia. Both Alzheimer’s and vascular dementia may increase the risk of ICH following thrombolytic therapy and, together, account for 80% of dementia cases [12].

Patients with and without dementia who were treated with thrombolysis for stroke have shown no difference in the incidence of haemorrhagic treatment complications in studies performed outside the UK [7,8,13]. However, the methodologies of these studies have some limitations by the nature of large-scale registry studies, such as including data not collected by a researcher and necessary data frequently being unavailable. Despite this, there is evidence that stroke patients with dementia have increased disability post-thrombolysis treatment [6,7]. Patients with cognitive impairment and ischaemic stroke may still benefit from thrombolytic therapy, although all the factors that lead to worse outcomes for this population of patients have not been identified.

Our study aimed to compare the thrombolysis outcomes of acute ischaemic stroke patients with and without pre-existing cognitive impairment and identify individual risk factors for poorer outcomes. This is the first UK-based study to attempt to answer this question and is a shift from the largely registry-based methodologies of previous studies.

## 2. Materials and Methods

This was a retrospective cohort study where the data were extracted from online and paper medical records. Ethical approval of the study protocol was obtained from the University of Sheffield Ethics Committee (Reference number: 038027), and the project was registered as a service evaluation with the Sheffield Teaching Hospitals NHS Trust, aimed at identifying factors that might improve the care of cognitively impaired patients on the acute stroke pathway.

### 2.1. Data Collection and Study Population Characterisation

The principles from the Declaration of Helsinki 1964 and the Data Protection Act 2018 were adhered to throughout the study. The study utilised paper medical records and the Electronic Document Management System (EDMS) and Integrated Clinical Environment (ICE) online databases as search engines. The data were collected from all the patients who attended the Sheffield Teaching Hospitals’ acute stroke pathway between January 2016 and February 2021 and received thrombolysis treatment. A time-based approach (<4.5 h from stroke onset) was used to guide the identification of patients eligible for thrombolysis. The patients with a final diagnosis of ischaemic stroke who were thrombolysed were included in this study. Patients were excluded where an ischaemic stroke was not diagnosed, or there was no evidence of thrombolysis treatment. Furthermore, those with inaccessible medical records were also excluded.

The patients were grouped into cognitively impaired and non-cognitively impaired cohorts (Table 1 describes the classification of cognitive impairment). The cognitively impaired cohort was further subdivided into three groups: a definite diagnosis of dementia, a definite diagnosis of mild cognitive impairment, and clinical evidence of cognitive impairment. The clinical evidence included pre-stroke cognitive assessment scores of <80 in the Addenbrooke’s Cognitive Examination-Revised and <25 in the Montreal Cognitive Assessment and Mini-Mental State Examination. These cut-off scores were validated for specificity and sensitivity, ensuring the reliability of their use as screening tools [14,15].

### 2.2. Outcome Measures

The outcome measures were categorised in terms of morbidity, mortality, and haemorrhagic complications. The National Institutes of Health Stroke Scale (NIHSS) scores on admission, post-thrombolysis treatment, and 24 h post-treatment were the measures of morbidity. The post-treatment scores were recorded 2–6 h following thrombolysis administration. Where 24 h NIHSS scores were not documented, the time frame for data acquisition was expanded up to 72 h. Premorbid and discharge Modified Rankin Scores (mRS) were recorded, with the latter being used as an outcome measure of morbidity. Haemorrhagic complications were categorised as asymptomatic, mild, or severe. The differentiation between asymptomatic and mild was based on the presence of symptoms, in addition to radiological evidence of haemorrhage. Severe haemorrhage was differentiated from mild when there was radiological evidence of a midline shift and mass effect in the brain parenchyma. An independent branch of classification was created for fatal haemorrhagic complications. This included patients whose cause of death as per the death certificate was “haemorrhagic transformation”, secondary to ischaemic stroke. Where no cause of death was recorded, patients with a severe haemorrhage who died during their initial admission or within 30 days of admission were assumed to have had a fatal haemorrhage. Regardless of the cause of death as per the death certificate, the patients were classified as having had a haemorrhage contributing to death if they had suffered from a haemorrhagic complication and died during admission or within 30 days. The mortality measures of interest included death during admission and within 90 days of admission.

### 2.3. Statistical Analysis

The statistical analysis was performed using IBM SPSS (version 26) and GraphPad Prism 9. A cohort demographic analysis was performed independently for all the subjects in the cognitively and non-cognitively impaired subgroups. Univariate comparisons of continuous variables between the groups were achieved using a t-test or Mann–Whitney U test, as appropriate to the distribution. Pearson’s chi-square test was performed to assess the statistical differences between the binary categorical variables. An odds ratio analysis for the incidence of all the outcome measures was calculated as a risk of incidence in the cognitively impaired cohort relative to the non-cognitively impaired cohort. The *p*-values were calculated according to logistic regression analysis.

Multivariate logistic regression analysis to determine the independent effect of cognitive impairment on each outcome measure was performed. Model 1 controlled for cognitive status alone. Model 2 controlled for cognitive status and variables which were significantly different between the control and cognitively impaired groups. Finally, Model 3 controlled for cognitive status and variables which were suspected to influence the risk of haemorrhagic complications. For all the analyses, a two-tailed *p*-value <0.05 was considered significant. 

## 3. Results

### 3.1. Demographic Data

Between January 2016 and February 2021, 478 patients at the study hospital were identified to have had an acute ischaemic stroke and been thrombolysed. The 478 patients were collected from a total of approximately 5000 patients who were seen with ischemic stroke in the study hospital during the same time frame. Of the 478 assessed for thrombolysis, 50 patients were excluded because, even though it was suggested they were thrombolysed, we found no evidence of this, or thrombolysis treatment was definitely not administered, or ischaemic stroke was not the final diagnosis. An alternative diagnosis of seizure was given to three patients. Of the 428 stroke patients who underwent thrombolysis, 62 were cognitively impaired. These patients were further subdivided into the following groups: 43 patients with a definite diagnosis of dementia, 14 with mild cognitive impairment, and 5 with clinical evidence of cognitive impairment (Figure 1).

The median age of the cognitively impaired stroke patients was higher than those without cognitive impairment (81 years (CI 76–90) and 76 years (CI 65.8–82), respectively, *p* < 0.001) (Table 2). The distribution of sex was equal in the non-cognitively impaired cohort. In contrast, the cognitively impaired cohort contained a significantly higher number of females (37, 59.7%) than males (25, 40.3%, *p* = 0.016) (Table 2). There was a trend towards a reduced time from symptom onset to the administration of thrombolysis in the non-cognitively impaired patients compared to those with cognitive impairment (181.5 min (CI 129.5–245.4) vs. 164.0 (CI 128.5–210), *p* = 0.074) (Table 2). In addition to thrombolysis, 46 (10.7%) patients also received mechanical thrombectomy treatment; all but two of these were non-cognitively impaired (Table 2).

Significant differences in the incidence of having a pre-existing diagnosis of AF, a history of previous stroke, and being a non-smoker were found between the cohorts. A greater proportion of cognitively impaired patients had a pre-existing diagnosis of AF (21, 33.9%) compared to 51 (13.9%) non-cognitively impaired patients, *p* < 0.001. In addition, cognitively impaired patients were more likely to have had a previous stroke (25, 40.3% vs. 56, 15.3%, *p* < 0.001). A total of 58 (93.5%) patients in the cognitively impaired cohort were non-smokers compared to 274 (74.9%) in the non-cognitively impaired cohort, *p* = 0.001 (Table 2).

### 3.2. Morbidity

As determined at admission, the cognitively impaired patients had more severe strokes than the non-cognitively impaired patients (median NIHSS 15 (CI 8–20) vs. 11 (CI 6–17.8), respectively, *p* = 0.015). Post-treatment and 24 h post-treatment, the NIHSS scores were, again, significantly higher in the cognitively impaired cohort (Table 3). Furthermore, the cognitively impaired patients were significantly less likely to show an improvement in the NIHSS score 24 h after thrombolysis (OR 0.48, CI 0.24–0.95, *p* = 0.035) (Table 3).

Pre-morbid mRS was higher in the cognitively impaired patients (mRS 3 (CI 2–3)) compared to the non-cognitively impaired patients (mRS 1, CI 0–1, *p* < 0.001) (Table 3, Figure 2). Similarly, at discharge, the cognitively impaired patients had a higher median mRS (4 (CI 3–6) vs. 2.5 (CI 1–4), *p* < 0.001) (Table 3). Although both groups were discharged with a greater disability, using a new mRS ≥ 4 at discharge outcome measure revealed 116 (32.0%) non-cognitively impaired patients and 31 (50.0%) cognitively impaired patients were discharged with this increased level of disability, suggesting that cognitively impaired patients present with worse morbidity, which then also gets worse at discharge (OR 2.01, CI 1.12–3.60, *p* = 0.008) (Table 3, Figure 2).

### 3.3. Mortality

Cognitively impaired patients were significantly more likely to die during admission (OR 3.46, CI 1.80–6.67, *p* < 0.001) or within 90 days (OR 3.34, CI 1.85–6.01, *p* < 0.001) (Table 3). Deaths during admission were most commonly due to pneumonia and haemorrhagic complications.

### 3.4. Haemorrhagic Complication

Despite a small increase in the incidence of haemorrhage reported in the cognitively impaired group (14 (22.6%) vs. 57 (15.6%)), this difference was not significant, *p* = 0.179. Cognitively impaired patients were over five times more likely to die following post-thrombolysis haemorrhage (OR 5.26, CI 1.56–17.82, *p* = 0.003) (Table 3). This suggests that, although the incidence of ICH post-thrombolysis is the same between the two groups, patients with cognitive impairment pre-thrombolysis are more likely to have a more severe post-thrombolysis haemorrhage if one occurs. There was no significant difference in the incidence of severe haemorrhage (OR 2.21, CI 0.94–5.19, *p* = 0.069) or post-thrombolysis haemorrhage which contributed to death (OR 2.32, CI 0.93–5.78, *p* = 0.063) between the two groups (Table 3).

### 3.5. Logistic Regression Analysis

Our initial analysis identified that the patients with cognitive impairment who had been thrombolysed were discharged with greater disability, had a higher risk of suffering haemorrhagic adverse events, and were less likely to survive post-thrombolysis compared to non-cognitively impaired patients. 

Binary logistic regression models were used to further analyse whether cognitive impairment independently predicted the poor outcomes observed, once the covariates had been controlled for. Model 1 controlled for cognitive impairment only. Model 2 controlled for factors significantly different between the two groups (age, sex, AF, previous stroke/TIA, smoking status), and model 3 controlled for factors suspected to impact the risk of haemorrhage (age, time to thrombolysis >120 min, NIHSS on admission (>6), and hypertension). 

In model 1, cognitive impairment was a significant predictor for death during admission and within 90 days (R^2^ = 0.054, *p* < 0.001 and R^2^ = 0.056, *p* < 0.001, respectively). The low Nagelkerke score demonstrates cognitive impairment only explains a small percentage of the total variability seen between the groups. Furthermore, cognitive status partially explains the increased number of new mRS ≥ 4 on discharge seen in this group (R^2^ = 0.023, *p* = 0.08) (Table 4).

In model 2, cognitive status was no longer a significant predictor of poor outcomes. The model predicted 26% of the variance in death within 90 days between the two groups, but this can be explained by the controlled-for factors, particularly age and a previous history of stroke or TIA (R^2^ = 0.260). Although following adjustments for covariates, the significance is lost, there is an analogous trend where cognitively impaired patients were more likely to die within admission (OR 1.91, CI 0.93–3.93, *p* = 0.080) or within 90 days (OR 1.56, CI 0.82–3.09, *p* = 0.171) (Table 4). A previous stroke or TIA was a significant factor to influence the risk of cognitively impaired patients dying within 90 days of admission (OR 2.07, *p* = 0.013), as was age (OR 1.11, *p* < 0.001) (Table 5). After controlling for age, sex, AF, previous stroke or TIA, and smoking status, cognitive impairment was no longer a significant predictor of a fatal haemorrhage (OR 3.39, CI 0.90–12.86, *p* = 0.072) (Table 4). No variables controlled-for in model 2 were significant contributors to explain the increased risk of fatal haemorrhages previously observed (Table 5).

Model 3 controlled for factors suspected to contribute to haemorrhagic complications. Controlling for age, time to thrombolysis, NIHSS on admission, and hypertension revealed cognitive impairment remained a significant predictor for death during admission (R^2^ 0.260, OR 2.21, CI 1.07–4.58, *p* = 0.033) and fatal haemorrhage (R^2^ 0.170, OR 4.79, CI 1.24–18.45, *p* = 0.023) between the groups (Table 4). In model 3, age was a significant predictor for the differences observed between the groups in the incidence of death during admission (OR 1.08, *p* < 0.001) and of haemorrhage contributing to death (OR 1.05, *p* = 0.046) (Table 5). 

Similar to the results seen in model 2, the independent factors controlled for in model 3 predicted a considerable percentage of the variance in survival outcomes between the groups. Cognitive status, time to thrombolysis, hypertension, age, and NIHHS score on admission collectively accounted for 26.5% of the variance observed between the groups for death during admission (R^2^ = 0.265). All the factors except hypertension were significant contributors to this outcome measure, including cognitive impairment (OR 2.21, *p* = 0.033) (Table 4).

In summary, after controlling for factors that were found to be significantly different between the cohorts, it appears that cognitive impairment does not account for a significant portion of the causation behind the worse outcomes observed in these patients. However, it does predict a four-fold increased risk of suffering a fatal haemorrhage, after controlling for factors thought to increase the risk of haemorrhage. 

## 4. Discussion

This was the first study in the UK to investigate the outcomes of cognitively impaired ischaemic stroke patients following thrombolytic therapy. Consistent with the current literature, we observed a greater severity of strokes and higher levels of pre-stroke disability in cognitively impaired patients [16]. Furthermore, we identified this group of patients to have poorer morbidity and mortality outcomes, in addition to a significantly increased risk of haemorrhagic complications post-thrombolysis. 

As in previous studies, we found cognitive impairment to be associated with longer hospital stays and increased post-discharge care needs following treatment [17]. The length of hospital stay is likely to be generally underestimated. Access to medical records from secondary hospitals, where over a quarter of patients (28.1% of non-cognitively impaired, 27.4% of cognitively impaired) continued their post-stroke care, was not available (Table 3). Admission NIHSS scores in cognitively impaired patients were higher, and this group was less likely to show an improvement in NIHSS scores following thrombolysis, which is a predictor of poor long-term outcomes [18]. 

Cognitively impaired patients with ischaemic stroke who were thrombolysed showed greater risks for death during the admission period and within 90 days of admission. The multimorbidity of these patients on admission is the main factor driving this; however, thrombolytic therapy may be exerting an influence. It is established that thrombolysis does not improve mortality following thrombolysis for acute ischaemic stroke. Its purported net benefit comes from improved functional outcomes on long-term follow-up [19,20,21,22]. Several studies have shown that, even when a patient presents with an mRS score of 2–4, they may still benefit from thrombolysis from a reduction in their post-stroke disability levels [23,24]. In a study of 15,317 thrombolysed patients by Gumbinger et al., the patients with a pre-thrombolysis mRS score of up to 4 still benefited from thrombolysis [23]. Additionally, Alshekhlee et al. found thrombolysis to be associated with higher rates of in-hospital mortality following ischaemic stroke [8]. This could be, in part, due to an increased risk of ICH following thrombolysis, which was reported in their study [8]. Therefore, although patients with cognitive impairment may be at higher risk of complications from thrombolytic therapy, a nuanced approach to selection for treatment is needed, as there may be an overall benefit.

The use of neuroimaging to aid diagnostics and guide treatment in strokes is increasingly being recognised as a valuable tool and can potentially aid in decision-making when a person has a pre-existing cognitive impairment. In our study centre, the decision to thrombolyse patients with acute ischaemic stroke primarily adopts a time-based approach, aiming to deliver treatment within 4.5 h. It has been suggested that using CT angiography and CT perfusion imaging to assess tissue viability are better predictors of good clinical outcomes following thrombolysis treatment [25]. 

Perfusion and previous MRI imaging may be of use in future studies to guide thrombolysis decisions and to identify potential the cerebral mechanisms which are responsible for poorer outcomes in cognitively impaired patients. Banerjee et al. noted that patients with pre-existing cognitive impairment are more likely to have previous cortical infarcts and lacunes. In addition, in these patients, MRI imaging revealed the presence of periventricular and deep white matter hyperintensities, caused by cerebral small vessel disease. Cerebral amyloid angiopathy, cerebral hypoperfusion, chronic inflammation, and endothelial dysfunction are mechanisms which may explain this association in patients with pre-existing cognitive impairment and poorer outcomes following thrombolysis treatment [26].

We found a significantly increased risk of fatal haemorrhagic complications in cognitively impaired ischaemic stroke patients following thrombolytic therapy. Underlying pathologies of cerebral vessel damage have been demonstrated to cause ICH in people with pre-existing dementia and MCI [27]. Thrombolytic therapy stimulates fibrin degradation, thereby increasing the risk of ICH, particularly where vessels are already frail and susceptible [28]. Due to the unavailability of radiological reports for all the patients, we used radiological evidence of midline shift and mass effect as per the clinical notes to categorise severe haemorrhage. While this has been demonstrated as a predictor of a poor outcome [29], the way we classified haemorrhage differs from the ECASS III criteria, which is used in the majority of previous studies [30]. This was due to the inconsistencies of the timings at which the post-thrombolysis NIHSS scores were recorded in the patient files and therefore limits the ability to compare our findings with those of previous studies. We classified a fatal haemorrhage based on whether it was recorded as the cause of death on the death certificate, regardless of the initial severity of the haemorrhage. Where no death certificate was available, the patients with a severe haemorrhage who died during admission or within 30 days were included. This applied to two patients from the cohort: One patient suffered from a severe haemorrhage with mass effect 5 days after thrombolysis treatment and was referred to coroners with the query of haemorrhagic transformation. For the other patient in which a fatal haemorrhage occurred, we were unable to find documentation of the death certificate; however, they suffered a significant ICH with intraventricular expansion and oedema, and subsequently died the day after thrombolysis. We observed that some patients who experienced severe haemorrhagic complications died shortly after treatment, but their death certificates attributed the cause of death to other complications such as pneumonia or sepsis. Nevertheless, it is reasonable to assume that the ICH played a significant role in their mortality and should be recognised. For this reason, we classified these patients as having a haemorrhage that contributed to their death. 

The presence of cognitive impairment appears to play a significant role in determining a fatal haemorrhage after thrombolysis treatment in stroke patients; nevertheless, we identify some caveats to this. In our previous analysis, a fatal haemorrhage was classified to include only patients who had died from ICH as per the death certification. Here, using identical logistic regression models, cognitive impairment was not a significant predictor of a fatal haemorrhage. This highlights the ongoing challenges in how we classify ICH and cognitive impairment, and the need for a larger, multi-centred, prospective study. 

Consistent with the findings of previous research [31,32,33], our logistic regression models confirmed that, in general, cognitive impairment is not a predictor of worse outcomes after thrombolysis. Death during admission and fatal haemorrhage were notable exceptions to this. Age, the NIHSS score on admission, time to thrombolysis, and a history of atrial fibrillation were the main predictors of poorer outcomes following thrombolysis. The higher occurrence of most of these risk factors among stroke patients with cognitive impairment provides an explanation for the poorer outcomes observed in this cohort. Nevertheless, the R^2^ values across all the logistic regression models were low, with the highest being 0.351. This suggests a significant proportion of the variance in outcome measures across the study cohort remains unexplained. This may be attributed to sample size limitations or unidentified risk factors that could contribute to worse outcomes. 

Our study has several notable strengths that are primarily due to our data collection and analysis methods. By directly obtaining data from medical records, we were able to circumvent the known limitations of registry-based studies, which have been detailed by Gallucio et al. [34]. We used anonymised and unique patient IDs during the data collection process, which eliminated the risk of duplicate records included in our study. Additionally, patient data were available to us through multiple sources, such as clinical notes, admission notes, stroke clerking proforma, and discharge summaries. This approach improved the quality of our collected data and allowed for cross-verification for greater reliability. Data from all the patients attending the stroke pathway who were assessed for thrombolysis were utilized in this study; exclusion criteria were limited to an absence of thrombolytic therapy or the lack of a confirmed diagnosis of stroke. Our low exclusion rate (10.5%) attests to this and indicates a minimal risk of selection bias.

The retrospective design of this study presented a limitation, as the patient information required for the analysis was not routinely recorded in each case. Consequently, there were some instances where documentation of the NIHSS and mRS were incomplete or recorded at different times post-thrombolysis. To address this issue, the data collection was performed by researchers with medical backgrounds. While they were able to estimate the mRS based on the clinical documentation, it has been suggested that this method of estimation can be inadequate due to interobserver variability in quantifying mRS [25]. The 24 h NIHSS post-thrombolysis outcome measure was extended to 72 h for the patients where 24 h scores were not recorded. In some cases, it may have been challenging for clinicians to identify cognitive impairment. The cognitive impairment could have been a secondary effect of stroke in some patients who presented acutely, therefore masking pre-existing impairment. Additionally, given the small sample sizes of the cognitively impaired subcategorization (five with clinical evidence of cognitive impairment and fourteen with mild cognitive impairment) (Figure 1), we were unable to perform a subset analysis. Moreover, certain comorbidities which are known to influence stroke severity, such as cancer [35], were not controlled for in the present study.

## 5. Conclusions

Stroke patients with cognitive impairments are at higher risk for morbidity, mortality, and haemorrhagic complications after undergoing thrombolytic therapy for stroke. Our study is the first to identify cognitive impairment in isolation as a significant predictor of a fatal haemorrhage, despite associated comorbidities also being strong influencers of outcomes. Some of the variability in the outcomes between our study groups is not fully explained. This emphasizes the need for a prospective study to control for other factors and monitor acute ischemic stroke patients with cognitive impairment who received thrombolytic therapy. Such research is crucial in informing clinical decision-making for these patients.

## Figures and Tables

**Figure 1 life-13-01055-f001:**
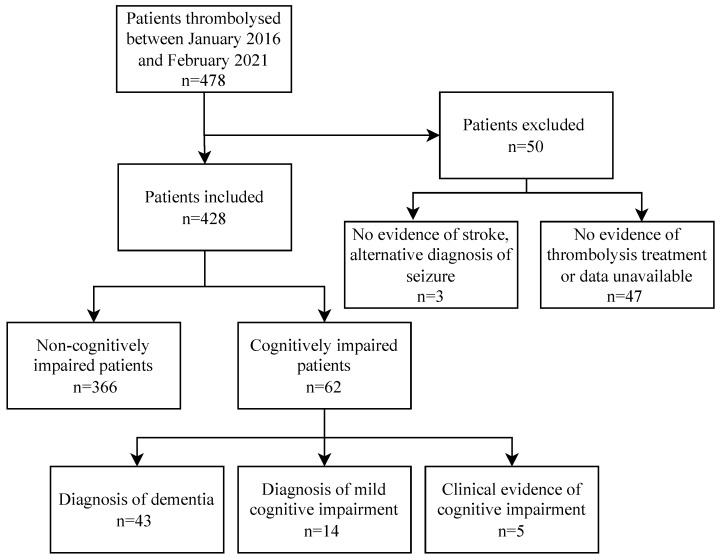
Displays the breakdown of cases reviewed in this cohort study.

**Figure 2 life-13-01055-f002:**
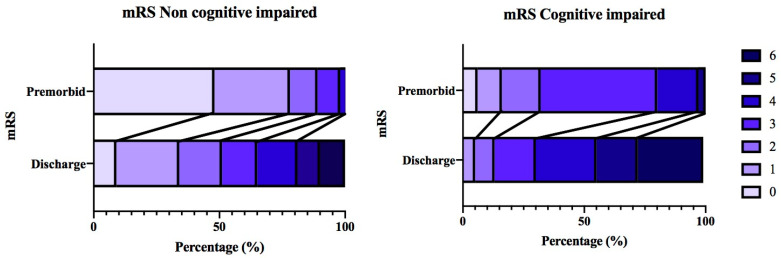
Displays morbidity outcome measures comparing the frequency distribution of premorbid and discharge mRS categorisations in the cognitively and non-cognitively impaired groups.

**Table 1 life-13-01055-t001:** Outlines the classification of how cognitive impairment was defined.

Type of Cognitive Impairment	
Definite diagnosis of dementia	Recorded pre-stroke diagnosis of dementia
Definite diagnosis of mild cognitive impairment	Recorded pre-stroke diagnosis of mild cognitive impairment
Clinical evidence of cognitive impairment	Recorded pre-stroke cognitive scores: <80 in the Addenbrooke’s Cognitive Examination-Revised <25 in the Montreal Cognitive Assessment <25 in the Mini-Mental State Examination

**Table 2 life-13-01055-t002:** Displays demographic data including age, time to thrombolysis, thrombectomy treatment, stroke location, and cardiovascular comorbidity. ^1^ indicates median + IQR, ^2^ indicates frequency and percentage of each group, and * indicates a statistically significant difference where *p* < 0.05. MCA = middle cerebral artery; AF = atrial fibrillation; MI = myocardial infarction; TIA = transient ischaemic attack; COPD = chronic obstructive pulmonary disease; CKD = chronic kidney disease.

	Non-Cognitively Impaired (n = 366)	Cognitively Impaired (n = 62)	*p*-Value
Age ^1^	76.0 (65.8–82.0)	81.0 (76.0–90.0)	<0.001 *
Time taken to thrombolysis (min) ^1^	164.0 (128.5–210.0)	181.5 (129.5–245.5)	0.074
Sex ^2^			
Male	208 (56.8%)	25 (40.3%)	0.016 *
Female	158 (43.2%)	37 (59.7%)	0.016 *
Thrombectomy ^2^	44 (12.0%)	2 (3.2%)	0.039 *
Stroke location ^2^			
Left MCA	134 (36.6%)	20 (32.3%)	0.509
Right MCA	126 (34.4%)	23 (37.1%)	0.683
Basilar	3 (0.8%)	1 (1.6%)	0.548
Other	99 (27.0%)	18 (29.0%)	0.746
Unknown	4 (1.1%)	0	0.408
Cardiovascular comorbidity (any) ^2^	324 (88.5%)	59 (95.2%)	0.104
New AF diagnosis	46 (12.6%)	7 (11.3%)	0.778
Pre-existing AF diagnosis	51(13.9%)	21 (33.9%)	<0.001 *
Hypertension	195 (53.3%)	32 (51.6%)	0.808
Previous MI	69 (18.9%)	16 (35.8%)	0.204
Previous stroke	56 (15.3%)	25 (40.3%)	<0.001 *
Previous TIA	30 (8.2%)	3 (4.8%)	0.359
Hyperlipidaemia	73 (19.9%)	10 (16.1%)	0.482
Non-smoker	274 (74.9%)	58 (93.5%)	0.001 *
Ex-smoker	42 (11.5%)	3 (4.8%)	0.115
Current smoker	50 (13.7%)	1 (1.6%)	0.007 *
Type 1 diabetes	3 (0.8%)	2 (3.2%)	0.103
Type 2 diabetes	59 (16.1%)	13 (21.0%)	0.345
Prediabetes	3 (0.8%)	0	0.474
COPD	37 (10.1%)	7 (11.3%)	0.777
CKD	39 (10.7%)	12 (19.4%)	0.054

**Table 3 life-13-01055-t003:** Displays outcome results following thrombolysis treatment in cognitively and non-cognitively impaired patients. Morbidity was evaluated using discharge location, NIHSS, and mRS scores. Mortality was evaluated using time-to-death during admission and within 90 days. Haemorrhagic complications were evaluated depending on severity. ^1^ indicates median + IQR, ^2^ indicates frequency and percentage of each group, ^3^ indicates odds of cognitively impaired against non-cognitively impaired, and * indicates a statistically significant difference where *p* < 0.05.

	Non-Cognitively Impaired (n = 366)	Cognitively Impaired (n = 62)	*p*-Value
Morbidity			
NIHSS on admission ^1^	11.0 (6.0–17.8)	15.0 (8.0–20.0)	0.015 *
NIHSS post-treatment ^1^	6.0 (3.0–12.0)	11.0 (6.0–19.0)	<0.001 *
NIHSS 24 h post-treatment ^1^	4.0.(1.0–10.0)	8.0 (4.0–17.0)	<0.001 *
Pre-morbid mRS ^1^	1.00 (0.0–1.0)	3.0 (2.0–3.0)	<0.001 *
Discharge mRS ^1^	2.5 (1.0–4.0)	4.0 (3.0–6.0)	<0.001 *
Discharge location			
Home ^2^	202 (55.2%)	14 (22.6%)	<0.001 *
Secondary care facility ^2^	103 (28.1%)	17 (27.4%)	0.907
Return to nursing home ^2^	8 (2.2%)	11 (17.7%)	<0.001 *
New admission to nursing home ^2^	12 (3.3%)	3 (4.8%)	0.539
Missing data ^2^	5 (1.4%)	0 (0%)	0.355
	**Odds ratio**	**95% CI**	** *p* ** **-value**
Discharge mRS > 2 ^3^	7.86	3.48–17.72	<0.001 *
Discharge mRS ≥ 4 ^3^	4.41	2.44–7.95	<0.001 *
New discharge mRS ≥ 4 ^3^	2.01	1.12–3.60	0.008 *
Post-treatment NIHSS improvement ^3^	0.57	0.31–1.06	0.074
24-h NIHSS improvement ^3^	0.48	0.24–0.95	0.035 *
Mortality			
Time to death from thrombolysis (days) ^2^	70.0 (19.0–466.0)	38.0 (10.5–277.5)	0.214
Death within admission ^2^	36 (9.8%)	17 (27.4%)	<0.001 *
	**Odds ratio**	**95% CI**	** *p* ** **-value**
Death during admission ^3^	3.46	1.80–6.67	<0.001 *
Death within 90 days ^3^	3.34	1.85–6.01	<0.001 *
Haemorrhagic complication			
Any haemorrhagic complication ^2^	57 (15.6%)	14 (22.6%)	0.176
Asymptomatic ^2^	17 (4.6%)	3 (4.8%)	0.947
Mild ^2^	17 (4.6%)	3 (4.8%)	0.947
Severe ^2^	23 (6.3%)	8 (12.9%)	0.069
Fatal ^2^	6 (1.6%)	5 (8.1%)	0.003 *
Haemorrhagic transformation contributing to death	19 (5.2%)	7 (11.3%)	0.063
	**Odds ratio**	**95% CI**	** *p* ** **-value**
Any haemorrhagic complication ^3^	1.58	0.82–3.05	0.176
Severe ^3^	2.21	0.94–5.19	0.069
Fatal ^3^	5.26	1.56–17.82	0.003 *
Haemorrhage contributing to death	2.32	0.93–5.78	0.063

**Table 4 life-13-01055-t004:** Shows binary logistic regression analysis for morbidity, thrombolysis complication, and survival outcomes. ^§^ Odds ratio, 95% confidence interval (CI), and *p*-value analysis presented predicts cognitive impairment as a variable of interest. * statistically significant difference, *p* < 0.05.

	R^2^ Nagelkerke	Odds Ratio ^§^	95% CI ^§^	*p*-Value ^§^
Model 1: Cognitive Impairment				
Morbidity				
Post-treatment NIHSS improvement	0.012	0.57	0.31–1.06	0.074
24 h NIHSS improvement	0.019	0.48	0.24–0.95	0.035 *
New mRS ≥ 4 on discharge	0.023	2.01	1.12–3.60	0.008 *
Mortality				
Death within admission	0.054	3.46	1.80–6.67	<0.001 *
Death within 90 days	0.056	3.34	1.85–6.01	<0.001 *
Haemorrhagic complication (any)	0.007	1.58	0.82–3.05	0.176
Severe haemorrhage	0.017	2.21	0.94–5.19	0.069
Fatal haemorrhage	0.068	5.26	1.56–17.82	0.003 *
Haemorrhage contributing to death	0.018	2.32	0.93–5.78	0.063
Model 2: Factors significantly different between study groups				
Morbidity				
Post-treatment NIHSS improvement	0.033	0.70	0.36–1.34	0.280
24 h NIHSS improvement	0.070	0.58	0.28–1.20	0.143
New mRS ≥ 4 on discharge	0.160	1.21	0.67–2.20	0.531
Mortality				
Death within admission	0.184	1.91	0.93–3.93	0.080
Death within 90 days	0.260	1.56	0.82–3.09	0.171
Haemorrhagic complication (any)	0.048	1.28	0.63–2.59	0.489
Severe haemorrhage	0.031	1.91	0.76–4.81	0.168
Fatal haemorrhage	0.139	3.39	0.90–12.86	0.072
Haemorrhage contributing to death	0.074	1.39	0.52–3.70	0.509
Model 3: Factors suspected to impact haemorrhagic complication				
Morbidity				
Post-treatment NIHSS improvement	0.059	0.59	0.31–1.15	0.120
24 h NIHSS improvement	0.058	0.64	0.31–1.31	0.219
New mRS ≥ 4 on discharge	0.240	1.33	0.73–2.43	0.357
Mortality				
Death within admission	0.260	2.21	1.07–4.58	0.033 *
Death within 90 days	0.351	1.73	0.86–3.45	0.123
Haemorrhagic complication (any)	0.057	1.29	0.65–2.59	0.469
Severe haemorrhage	0.058	2.17	0.86–5.43	0.099
Fatal haemorrhage	0.170	4.79	1.24–18.45	0.023 *
Haemorrhage contributing to death	0.152	1.61	0.60–4.30	0.340

**Table 5 life-13-01055-t005:** Shows binary logistic regression model 2 controlled for sex, AF, previous history of stroke or TIA, and smoking status, and model 3 controlled for +time to thrombolysis (>120 min), hypertension, age, and NIHHS score on admission >6. Only variables which were significant predictors (*p* < 0.05) of the outcome are presented with the odds ratio (OR) calculated as cognitively vs. non-cognitively impaired group.

	Model 2: Covariates Showing Significant Correlation		Model 3: Factors Suspected to Impact Haemorrhagic Complications	
	Covariates Showing Significant Correlation	R^2^ Nagelkerke	Covariates Showing Significant Correlation	R^2^ Nagelkerke
Morbidity				
Post-treatment NIHSS improvement	No significant variables	0.033	Time to thrombolysis: OR 0.42, *p* = 0.019	0.059
24 h NIHSS improvement	Age: OR 0.96, *p* = 0.007	0.070	Age: OR 0.97, *p* = 0.027	0.058
New mRS ≥ 4 on discharge	Age: OR 1.05, *p* < 0.001 AF: OR 2.26, *p* = 0.001	0.160	Age: OR 1.04, *p* < 0.011 Admission NIHSS > 6: OR 7.21, *p* < 0.001	0.240
Mortality				
Death within admission	Age: OR 1.08, *p* < 0.001 AF: OR 1.96, *p* < 0.036	0.184	Cognitive status: OR 2.21, *p* = 0.033 Age: OR 1.08, *p* < 0.001 Time to thrombolysis: OR 3.35, *p* = 0.031 Admission NIHSS > 6: OR 19.0, *p* = 0.004	0.260
Death within 90 days admission	Age: OR 1.11, *p* < 0.001 Prev stroke/TIA: OR 2.07, *p* = 0.013	0.260	Age: OR 1.11, *p* < 0.001 Time to thrombolysis: OR 3.85, *p* = 0.005 Admission NIHSS > 6: OR 16.9, *p* < 0.001	0.351
Haemorrhagic complication (any)	Smoking: OR 0.43, *p* = 0.036	0.048	Admission NIHSS > 6: OR 2.98, *p* = 0.006	0.057
Severe haemorrhage	No significant variables	0.031	No significant variables	0.058
Fatal haemorrhage	No significant variables	0.139	Cognitive status: OR 4.79, *p* = 0.023	0.170
Haemorrhage contributing to death	No significant variables	0.074	Age: OR 1.05, *p* = 0.046	0.152

## Data Availability

The data presented in this study are available on request from the corresponding author. The data are not publicly available due to privacy regulations.

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
