# Peer review of "Thrombolysis Outcomes in Acute Ischaemic Stroke Patients with Pre-Existing Cognitive Impairment"

_life, 2023, doi:10.3390/life13041055_

Round 1
Reviewer 1 Report
This manuscript presents interesting retrospective data regarding the differences in clinical and functional outcomes of people who received thrombolytic therapy after an acute ischemic stroke event, according to preexisting cognitive impairment. The findings are interesting and worth consideration.
I would just ask what were the criteria adopted for tPA (they only describe in the introduction that it usually administered before 4.5 h); indeed, recent literature suggests that a tissue-based approach (imaging based) instead of a time-based approach could help to better identify those who might benefit from the thrombolytic therapy.
Do the authors have any data about imaging (as CTP or MRI), or EEG, as they might help to predict the outcome and also might be related to pre-existing conditions, including dementia/cognitive impairment (10.1161/JAHA.119.014537; 10.1007/s11517-020-02280-z). I think that it might interesting to discuss it, as it might help to suggest the potential mechanisms associated with worse outcomes in stroke patients with pre-existing cognitive impairment.
Author Response
This manuscript presents interesting retrospective data regarding the differences in clinical and functional outcomes of people who received thrombolytic therapy after an acute ischemic stroke event, according to preexisting cognitive impairment. The findings are interesting and worth consideration.
I would just ask what were the criteria adopted for tPA (they only describe in the introduction that it usually administered before 4.5 h); indeed, recent literature suggests that a tissue-based approach (imaging based) instead of a time-based approach could help to better identify those who might benefit from the thrombolytic therapy.
Do the authors have any data about imaging (as CTP or MRI), or EEG, as they might help to predict the outcome and also might be related to pre-existing conditions, including dementia/cognitive impairment (10.1161/JAHA.119.014537; 10.1007/s11517-020-02280-z). I think that it might interesting to discuss it, as it might help to suggest the potential mechanisms associated with worse outcomes in stroke patients with pre-existing cognitive impairment.
We thank the reviewers for their comments. The criteria adopted for tPA was time-based. The following has been added to the methods section to make this clear:
“A time-based approach (<4.5 hours from stroke onset) was used to guide identification of patients eligible for thrombolysis”
In our study centre, CT perfusion and MRI imaging pre-thrombolysis are very rare, therefore we do not have data which would add to our analysis about the use of imaging to predict outcome. We acknowledge this is interesting to discuss and have added the following to the discussion section, paragraph 4 and 5 of the manuscript:
“The use of neuroimaging to aid diagnostics and guide treatment in stroke is increasingly being recognised as a valuable tool and can potentially aid in decision making when a person has a pre-existing cognitive impairment. In our study centre, the decision to thrombolyse patients with acute ischaemic stroke primarily adopts a time-based approach, aiming to deliver treatment within 4.5 hours. It has been suggested that using CT angiography and CT perfusion imaging to assess tissue viability, are better predictors of good clinical outcome following thrombolysis treatment [25].
“Perfusion and previous MRI imaging would be of use in future studies, to guide thrombolysis decisions, and to identify potential cerebral mechanisms which are responsible for poorer outcome in cognitively impaired patients. Banerjee et al noted that patients with pre-existing cognitive impairment are more likely to have previous cortical infarcts and lacunes. In addition, in these patients, MRI imaging revealed the presence of periventricular and deep white matter hyperintensities, caused by cerebral small vessel disease. Cerebral amyloid angiopathy, cerebral hypoperfusion, chronic inflammation, and endothelial dysfunction are mechanisms which may explain this association in patients with pre-existing cognitive impairment and poorer outcomes following thrombolysis treatment [26].”
Reviewer 2 Report
The authors of this well designed and written study aimed to compare the thrombolysis outcomes of acute ischaemic stroke patients with and without pre-existing cognitive impairment.
The results suggest that cognitive impairment is a relevant risk factor for fatal hemorrhage after thrombolysis, adding some data to the important issue of treating with reperfusive therapies patients with pre-morbid disability.
Authors wisely did not state that reperfusive therapies should be avoided in cognitively impaired patients.
I would like authors to add more studies that tried to address the feature of reperfusive patients in patients with pre morbid disability and cognitive impairment as:
Caruso P, Ajčević M, Furlanis G, Ridolfi M, Lugnan C, Cillotto T, Naccarato M, Manganotti P. Thrombolysis safety and effectiveness in acute ischemic stroke patients with pre-morbid disability.J Clin Neurosci. 2020; 72:180–184. doi: 10.1016/j.jocn.2019.11.047
Gumbinger C, Ringleb P, Ippen F, Ungerer M, Reuter B, Bruder I, Daffertshofer M, Stock C; Stroke Working Group of Baden-Württemberg. Outcomes of patients with stroke treated with thrombolysis according to prestroke Rankin Scale scores.Neurology. 2019; 93:e1834–e1843. doi: 10.1212/WNL.0000000000008468
Author Response
The authors of this well designed and written study aimed to compare the thrombolysis outcomes of acute ischaemic stroke patients with and without pre-existing cognitive impairment.
The results suggest that cognitive impairment is a relevant risk factor for fatal hemorrhage after thrombolysis, adding some data to the important issue of treating with reperfusive therapies patients with pre-morbid disability.
Authors wisely did not state that reperfusive therapies should be avoided in cognitively impaired patients.
I would like authors to add more studies that tried to address the feature of reperfusive patients in patients with pre morbid disability and cognitive impairment as:
Caruso P, Ajčević M, Furlanis G, Ridolfi M, Lugnan C, Cillotto T, Naccarato M, Manganotti P. Thrombolysis safety and effectiveness in acute ischemic stroke patients with pre-morbid disability.J Clin Neurosci. 2020; 72:180–184. doi: 10.1016/j.jocn.2019.11.047
Gumbinger C, Ringleb P, Ippen F, Ungerer M, Reuter B, Bruder I, Daffertshofer M, Stock C; Stroke Working Group of Baden-Württemberg. Outcomes of patients with stroke treated with thrombolysis according to prestroke Rankin Scale scores.Neurology. 2019; 93:e1834–e1843. doi: 10.1212/WNL.0000000000008468
We thank the reviewer for their comments and have added a reference to the above studies in the discussion of the paper, preceded by the following text:
“Several studies have shown that even when a patient presents with a mRS score of 2-4 they may still receive benefit from thrombolysis by reducing post stroke disability levels [23, 24]. In a study of 15,317 thrombolysed patients by Gumbinger et al, patients with a pre-thrombolysis mRS score of up to 4 still benefited from thrombolysis [23].”
Reviewer 3 Report
The paper by Ramnarine et al deals with an interesting topic, namely the impact of a previous cognitive impairment on the outcome of thrombolysis, whether or not cognitive impairment was defined.
The paper is interesting. However I see some points that should be clarified.
Some of the methods are not fully explained. For example, what does it mean that NIHSS and MRS were inconsistent? In methods authors explained a contingency strategy to limit this bias but probably it didn’t work. Please, expand.
As first results, I would point out how larvae was the original sample of all ischemic stroke patients in the same time interval to better understand to what the 10% of selection bias is referred to.
I would recommend authors to use a multiple comparison correction to trim the results from the univariate. The main contesting result of this study is that cognitive impairment alone does not predict a poorer outcome but a higher rate of hemorrhage transformation. This is puzzling. And it could be explained by the fact that too many covariates were inserted in the logistic models 2 and 3. So I would be curious to see what happens with just multiple comparison corrected results. Otherwise the choice to considered hemorrhage transformation as the cause of death only on the temporal link could be uncorrected. I would like to know how many patients had the hemorrhage transformation registered as primary cause of death in the death certificate and in how many this datum was derived. For this latter percentage I would like to know which were the other following complications that led to death.P
Author Response
The paper by Ramnarine et al deals with an interesting topic, namely the impact of a previous cognitive impairment on the outcome of thrombolysis, whether or not cognitive impairment was defined.
The paper is interesting. However I see some points that should be clarified.
Some of the methods are not fully explained. For example, what does it mean that NIHSS and MRS were inconsistent? In methods authors explained a contingency strategy to limit this bias but probably it didn’t work. Please, expand.
We thank the reviewers for their comments NIHSS and mRS inconsistencies from data collection addresses the limitation observed during data collection where NIHSS scores were not always documented at exactly 2 hours and 24 hours after thrombolysis. Therefore, we expanded the timeframes to encompass the data available to us for the majority of patient notes (ie 2-6 hours after thrombolysis and 72 hours after thrombolysis). mRS estimation by researchers is suggested as a potential limitation in this study due to findings from Quinn et all [31] who suggest the subjective quantification of scores and the interobserver variability of calculating mRS. This has been made clear in the preceding text in the methods section 2.2 outcome measures and in the last paragraph of the discussion of the manuscript.
“National Institutes of Health Stroke Scale (NIHSS) scores on admission, post-thrombolysis treatment, and 24 hours post-treatment were measures of morbidity. Post-treatment scores were recorded 2-6 hours following thrombolysis administration. Where 24-hour NIHSS scores were not documented, the time frame for data acquisition was expanded up to 72 hours.”
“The retrospective design of this study presented a limitation as patient information required for the analysis was not routinely recorded in each case. Consequently, there were some instances where documentation of NIHSS and mRS were incomplete or recorded at different times post-thrombolysis. To address this issue, data collection was performed by researchers with medical backgrounds. While they were able to estimate the mRS based on the clinical documentation, it has been suggested that this method of estimation can be inadequate due to interobserver variability of quantifying mRS [31]. The 24-hour NIHSS post-thrombolysis outcome measure was extended to 72 hours for patients where 24-hour scores were not recorded.”
As first results, I would point out how larvae was the original sample of all ischemic stroke patients in the same time interval to better understand to what the 10% of selection bias is referred to.
We thank the reviewer for their comment, and apologise for the confusion. The 10% selection bias number refers to all patients that where thought to have thrombolysis during the study period, not all patients that where assessed in the stroke unit. We have clarified this statement in the discussion with the following text.
“Data from all patients attending the stroke pathway who were assessed for thrombolysis was….”
We have also added the approximate number of all ischemic strokes seen within our unit over the same study period to the initial results section paragraph. This was approximately 5000.
I would recommend authors to use a multiple comparison correction to trim the results from the univariate. The main contesting result of this study is that cognitive impairment alone does not predict a poorer outcome but a higher rate of hemorrhage transformation. This is puzzling. And it could be explained by the fact that too many covariates were inserted in the logistic models 2 and 3. So I would be curious to see what happens with just multiple comparison corrected results. Otherwise the choice to considered hemorrhage transformation as the cause of death only on the temporal link could be uncorrected.
We thank the reviewer for their comments, in model 1 we show that cognitive impairment is a significant predictor of poorer outcome, we have clarified the text to make this point clearer.
The results from the regression models have been presented with unadjusted p-values, as the study's aim was "to compare and explore factors contributing to thrombolysis outcome", not to conclusively establish a significant association based on a pre-specified hypothesis. The reader is free to apply their preferred multiple comparison correction, for example by applying an alternative significance threshold such as 0.05 divided by the number of tests presented. For example, using 0.05/9 = 0.0056 for the 9 tests presented for Model 1 (Table 4), the results for both of the mortality endpoints and "fatal haemorhage" would remain significant. However, as demonstrated by the results of models 2 and 3, this apparent effect is at least partly driven by imbalances between groups with respect to age and cardiovascular comorbidity.
I would like to know how many patients had the hemorrhage transformation registered as primary cause of death in the death certificate and in how many this datum was derived. For this latter percentage I would like to know which were the other following complications that led to death.
We thank the reviewers for their comments, and we would like to clarify that fatal haemorrhage occurred in 2/11 patients. The paragraph below has been added to the discussion of the manuscript:
“This applied to 2 patients from the cohort; one patient suffered from a severe haemorrhage with mass effect 5 days after thrombolysis treatment and were referred to coroners with the query of haemorrhagic transformation. For the other patient in which fatal haemorrhage occurred, we were unable to find documentation of the death certificate, however, they suffered a significant ICH with intraventricular expansion and oedema, and subsequently died the day after thrombolysis.”
Round 2
Reviewer 3 Report
Authors successfully replied to my comments. I endorse the publication.